# Human Adipose Tissue-Derived Stromal Cells Suppress Human, but Not Murine Lymphocyte Proliferation, via Indoleamine 2,3-Dioxygenase Activity

**DOI:** 10.3390/cells9112419

**Published:** 2020-11-05

**Authors:** Adriana Torres Crigna, Stefanie Uhlig, Susanne Elvers-Hornung, Harald Klüter, Karen Bieback

**Affiliations:** 1Medical Faculty Mannheim, Institute of Transfusion Medicine and Immunology, Heidelberg University, German Red Cross Blood Service Baden-Württemberg-Hessen, 68167 Mannheim, Germany; Adriana.Torres_Crigna@ukbonn.de (A.T.C.); susanne.elvers-hornung@medma.uni-heidelberg.de (S.E.-H.); harald.klueter@medma.uni-heidelberg.de (H.K.); 2FlowCore Mannheim Medical Faculty Mannheim, Heidelberg University, 68167 Mannheim, Germany; stefanie.uhlig@medma.uni-heidelberg.de; 3Medical Faculty Mannheim, Mannheim Institute for Innate Immunoscience, Heidelberg University, 68167 Mannheim, Germany

**Keywords:** mesenchymal stromal cells, immunomodulation, IDO, extracellular vesicles, allogeneic, xenogeneic

## Abstract

Over recent years, mesenchymal stromal cells (MSC) have gained immense attraction in immunotherapy, regenerative medicine and tissue engineering. MSC microenvironment modulation occurs through synergy of direct cell–cell contact, and secreted soluble factors and extracellular vesicles (EV). MSC-derived EV have been suggested as cell-free immunomodulatory alternative to MSC; however, previous findings have challenged this. Furthermore, recent data suggest that evaluating the mechanism of action of human MSC (hMSC) in animal models might promote adverse immune reactions or lack of functionality due to xeno-incompatibilities. In this study, we first assessed the immunomodulatory strength of different human MSC sources on in vitro stimulated T cells and compared this to interferon-gamma (IFNγ) primed MSC conditioned medium (CM) and EV. Second, we addressed the main molecular mechanisms, and third, we assessed the MSC in vitro immunosuppressive effect across interspecies barriers. We identified human adipose tissue-derived stromal cells (ASC) with strongest immunomodulatory strength, followed by bone marrow (BM) and cord blood-derived MSC (CB). Whilst CM from primed ASC managed to exert analogous effects as their cellular counterpart, EV derived thereof did not, reproducing previous findings. IFNγ-induced indoleamine 2,3-dioxygenase (IDO) activity was identified as key mechanism to suppress human lymphocyte proliferation, as in the presence of the IDO inhibitor epacadostat (Epac) a stimulation of proliferation was seen. In addition, we revealed MSC immunosuppressive effects to be species-specific, because human cells failed to suppress murine lymphocyte proliferation. In summary, ASC were the strongest immunomodulators with the IDO-kynurenine pathway being key within the human system. Importantly, the in vitro lack of interspecies immunomodulatory strength suggests that preclinical data need to be carefully interpreted especially when considering a possible translation to clinical field.

## 1. Introduction

Multipotent mesenchymal stromal cells (MSC) are mesoderm derived, fibroblast-like cells when cultured in vitro. They differentiate into numerous mesenchymal lineages and have been isolated from a variety of sources such as adipose-tissue (ASC), bone marrow (BM-MSC), umbilical cord tissue (UC) or cord blood (cord blood-derived MSC—CB-MSC) [1,2]. According to their source they may display not only phenotypical but also multiple functional differences [3,4]. However, studies where their therapeutic benefits are directly compared, are limited.

MSC interest as therapeutic agents has increased, as they possess low immunogenicity but broad immunomodulatory potential. MSC can modulate both innate and adaptive immune responses in numerous diseases, acting on any immune cell type to counteract adverse inflammatory reactions and balance them towards pro-regenerative responses.

MSC microenvironment modulation occurs through synergy of direct cell-cell contact, such as programmed death ligand-1 (PDL-1), in cooperation with secreted soluble factors and released extracellular vesicles (EV) [5]. Plenty of factors can be involved, including indoleamine 2,3-dioxygenase (IDO), hepatocyte growth factor (HGF) and nitric oxide (NO) [6]. However, it is still not understood in detail yet, which factors are mostly involved in which particular setting.

It appears that MSC need to be “licensed” or “primed” to exert their paracrine function optimally, which can be attained by several different external stimuli [7]. MSC priming with IFNγ induces expression of IDO, known to be immunomodulatory [8]. IDO increased secretion and further breaking down of essential amino acid tryptophan (Tryp) has been closely related to MSC T cell inhibitory potentials. Tryptophan depletion, along with an accumulation of metabolites along the kynurenine (Kyn) pathway may lead to a tolerogenic milieu [9,10]. In these terms, ASC, BM and CB appear to modulate lymphocyte proliferation via IDO to a large extent [8]. Furthermore, in presence of proinflammatory cytokines, inducible nitric oxide synthase (iNOS) stimulate secretion of NO leading to inhibition of T cell proliferation [11]. In vivo and in vitro investigations have reported that iNOS deficiency in murine MSC results in decreased inhibitory capacities [12]. Indeed, MSC inhibitory mechanisms differ amongst different species. While the IDO-Kyn pathway is relevant in human, nitric oxide synthase (NOS) produced NO appears to be most relevant in rat/mouse MSC with limited IDO action [13].

With increasing knowledge, it became evident that some of the regulatory activities relate to soluble factors and thus are found in MSC-CM and EV [14]. EV anti-apoptotic, angiogenic and regenerative potentials have made them rise as a potential alternative cell-free therapeutic prospect circumventing many of the safety challenges regarded in cell therapy [15]. EV, small vesicles secreted by a variety of cell types, are known to take part in cell-cell communication [16]. Discrepant results regarding EV immunomodulatory properties prompted us to re-investigate on this [17,18,19,20,21,22,23].

MSC are widely considered for treatment as cell-based therapies; however, the use of human MSC (hMSC) in animal models gives rise to increasing debates and concerns. The possibility of an adverse immune reaction or even the non-functionality of certain modes of actions represents an important aspect that cannot be underestimated when considering animal models for assessing therapeutic effects of human MSC. Cross species function studies are inconsistent. On one hand, multiple xenogeneic experimental models have demonstrated positive immunomodulatory function. Moreover, hASC implanted into immunocompetent rat hearts in a murine model of myocardial infarction were able to survive the xenogeneic mismatch for an extended period, revealing significant cardiac function amelioration [24]. Conversely, some studies reported humoral and/or cellular responses after xenogeneic MSC administration [25]. Additionally, in a recent study conducted by Lohan et al., immunomodulation of hMSC was ineffective in a rat model of corneal transplantation due to interspecies incompatibilities. Indeed, the authors claim hMSC are not able to be properly primed in a xenogeneic environment and therefore cannot exert their immune modulatory action [26].

Collectively, these data clearly indicate pronounced discordances regarding the outcome of xenogeneic MSC infusion. Overall, it is challenging to guarantee hMSC long-term engraftment or functionality in animals.

The approach of our study was to perform in vitro experiments to identify: (a) the immunomodulatory potential of different human MSC sources on in vitro stimulated T cells, (b) compare their strength to MSC secreted CM and EV and (c) establish the main molecular mechanisms involved. Furthermore, we aimed to assess: (d) MSC in vitro immunosuppressive effect across interspecies barriers. In summary, our results suggest that IDO-kynurenine pathway is essential for human MSC-mediated T cell suppression and while CM from primed MSC exert comparable immunomodulation than their parental cells, EV lack suppressive potential. Furthermore, human ASC lack suppressive potential in a xenogeneic in vitro system using murine lymphocytes.

## 2. Materials and Methods

### 2.1. Cell Culture

Bone marrow (BM), cord blood (CB) and adipose derived mesenchymal stromal cells (ASC) were isolated from numerous healthy donors after obtaining informed consent. The study was approved by the Mannheim Ethics Commission II (vote numbers 2015-520N-MA, 49/05 and 2006-192 N-MA, respectively). BM, CB and ASC were isolated and characterized as previously described [27]. Following isolation, ASC and BM were continuously cultured at a density of 200 cells/cm^2^ and CB at 700 cells/cm^2^ in DMEM (PAN Biotech; Aidenbach, Germany) supplemented with 10% pooled human allogeneic AB serum from healthy donors (German Red Cross Blood Donor Service Baden-Württemberg—Hessen, Mannheim, Germany), 1% penicillin/streptomycin (100,000 U/mL penicillin and 10 mg/mL streptomycin; PAN Biotech) and 2% L-glutamine (200 mM; PAN Biotech) (=DMEM-AB). Cells were cultured in incubators at 37 °C and 5% CO_2_.

Rat bone marrow derived MSC (rMSC) were isolated from male Sprague-Dawley (SD) rat femurs [28] and underwent FACS characterisation. All rats used in these experiments were control animals from other animal experiments with their corresponding animal experimentation permissions. rMSC were continuously cultured at a density of 200 cells/cm^2^ in DMEM (PAN Biotech) supplemented with 10% foetal bovine serum (FBS) (Thermo Fisher Scientific; Waltham, MA, USA), 1% penicillin/streptomycin (PAN Biotech), 2% L-glutamine (200 mM; PAN Biotech) (=DMEM-FBS). rMSC immunophenotype was assured with the following anti-rat antibodies: anti-CD44-APC (Clone 12K35; Seattle, WA, USA), anti-CD45-FITC (Clone OX-1; Bio-Rad; Feldkirchen, Germany) to exclude hematopoietic cells, anti-CD90-PE (Clone OX-7; BD Biosciences, Heidelberg, Germany). Upon reaching confluency, MSC and rMSC were trypsinised with 1× trypsin/EDTA (PAN Biotech), counted and seeded according to the experiment. Cells were cryopreserved in FBS with 10% DMSO (Wak-chemie Medical GmbH; Steinback, Germany) and were always thawed and cultured for at least one passage prior to their use in experiments. Cell growth and morphology were monitored by microscope observation (AxioVert100 Zeiss; Oberkochen, Germany).

### 2.2. Peripheral Blood Mononuclear Cell Isolation

Human peripheral blood mononuclear cells (hPBMC) were isolated from either buffy coats or leukapheresis samples from healthy donors provided to us by the German Red Cross Blood Donor Service Mannheim. PBMC were obtained after performing Ficoll-Paque™ (GE Healthcare; Uppsala, Sweden) density gradient isolation and ammonium chloride erythrocyte lysis. PBMC pellets were resuspended in PBS-EDTA (2 mM; AppliChem; Darmstadt, Germany). After isolation, cell numbers were determined and PBMC were cryopreserved. To compare the inhibitory effect of MSC on whole PBMC population against enriched CD4+ T cells. CD4+ T cells were enriched from the initial whole PBMC population by using CD4+ T cell isolation kit (Miltenyi Biotec; Bergisch-Gladbach, Germany) following the manufacturer’s instructions.

Rat blood-derived PBMC (rPBMC) were isolated from freshly collected blood of SD rats. PBMC density gradient isolation was performed as previously described. Spleen mononuclear cells (SMC) were isolated from SD rat spleens immediately after dissection. Spleens were cut finely into small sections and put onto a 100 µm and 70 µm cell strainer in a PBS-antibiotic/antimycotic solution and then centrifuged. Pellets were resuspended in 1X Erythrocyte lysis buffer. After centrifugation pellets were resuspended in PBS-antibiotic/antimycotic (GE Healthcare; Uppsala, Sweden). Cell numbers were determined and cells were freshly used.

### 2.3. Conditioned Media Preparation

ASC-derived CM was prepared by seeding 1.5 × 10^6^ MSC in a T175 flask, or for EV production in an 875 cm^2^ 5-layer multiflask (Falcon, Fischer Scientific; Schwerte, Germany), in full DMEM AB. Cells were allowed to attach overnight. The next day, media was changed to 20 mL RPMI 1640 supplemented with 10% FBS, 1% Penicillin/Streptomycin and 2% L-glutamine (200 mM) (=RPMI). For EV production, DMEM with 10% AB serum was used as it promoted high cell expansion within the multiflasks with improved EV yields. To avoid contamination with serum EV, the AB serum was previously ultracentrifuged (for detailed protocol see Appendix A). ASC were primed with IFNγ (final concentration 10 ng/mL, R&D Systems, Minneapolis, MN, USA) or left untreated. CM was left conditioning for 72 h, after which CM was collected and stored at −30 °C for use in experiments or proceeded with EV isolation procedure. To calculate the number of producer cells present at the moment of harvesting, cells were washed thoroughly with PBS, trypsinised and counted. CM and EV were always calculated to the number of producer cells and applied as cell equivalents. An EV media control was constituted in the same manner as MSC-derived CM although generated in absence of MSC.

### 2.4. EV Isolation and Characterisation

EV isolation and purification was performed by consecutive steps of differential centrifugation. CM was collected and after one first centrifugation step, tubes were placed in a WX Ultra Series 100 ultracentrifuge for a first UC step of 10,000× *g* and a second UC step of 105,000× *g* of 45 min at 10 °C [29]. EV pellets were resuspended in sterile filtered PBS and adjusted to yield 200 µL per 2 × 10^7^ producer cells and stored in low adhesive tubes (Biozym Scientific; Hessisch Oldendorf, Germany) at −30 °C for a maximum of 6 months.

EV characterisation was performed according to nanotracking analysis measurement (NTA), transmission electron microscopy (TEM), and flow cytometry detection. For detailed isolation and characterisation protocols see Appendix A.

### 2.5. PBMC Proliferation Assay

#### 2.5.1. Cytotell Green Proliferation Dye

To assess T cell proliferation, a minimum of 4 × 10^7^ PBMC (human and rat) or human CD4+ T cells were resuspended in PBS and stained with the proliferation dye Cytotell Green, which allows to monitor cell division over time due to its uniformly distribution amongst daughter cells in each division (ATT Bioquest; Sunnyvale, CA, USA) (final concentration 1:500 dilution from company stock). After 15 min incubation at 37 °C, cells were washed, centrifuged and resuspended in RPMI and seeded appropriately.

#### 2.5.2. Mitogen Stimulation

hPBMC were left unstimulated or stimulated with phytohemagglutinin-L (PHA) (PHA-L pure, Biochrom, Merck Millipore; Darmstadt, Germany) (1.25 µg/mL) and IL-2 (11 μg/mL, Promocell; Heidelberg, Germany). rPBMC and rSMC were left unstimulated or stimulated with concanavalin A (ConA; 4 µg/mL final concentration, Sigma-Aldrich, Merck KGaA, Darmstadt, Germany) and β-mercaptoethanol (β-ME; 50 µM, Sigma-Aldrich).

#### 2.5.3. Coculture Setup

Different ratios of MSC:PBMC or CD4+ T cells labelled with Cytotell Green were seeded for the cocultures (1 × 10^5^ PBMC/CD4+ T cells). Whole PBMC population was compared to enriched CD4+ T cells to establish if presence of accessory cells are imperative for suppression by MSC [30]. Cells were added either directly on top of the MSC (direct coculture system) or in a transwell insert (0.4 µm polyethylene terephthalate (PET) membrane; Falcon, Fischer Scientific; Schwerte, Germany). According to the experiments, tryptophan (final concentration 100 µg/mL; Santa Cruz Biotechnology; Heidelberg, Germany) or IDO inhibitor epacadostat (Epac; final concentration 1 µM; Selleckchem; Munich, Germany) were added. Instead of using MSC, CM (volume, equivalent to a 1:5 MSC:PBMC ratio) and EV (originated from 2 × 10^6^ cells, equivalent to a 20:1 MSC:PBMC ratio) was added. PBMC, stimulated and not stimulated with mitogen, were seeded as controls in the absence of MSC. Both direct and indirect cocultures were set in parallel for comparison purposes. After 5 days, cocultures were harvested, and CM was collected for further testing. To investigate the potential inhibitory effect of CM on PBMC proliferation, another set of cocultures were performed with CM harvested from previous cocultures (5 days) that was diluted 1:2 in new RPMI medium in which newly thawed PBMC were resuspended and seeded.

Rat:human MSC:PBMC/SMC cocultures ran in a similar manner as described above. A series of xeno- and allo-cocultures were investigated. These were xeno-cocultures: (a) hMSC + rPBMC/rSMC, 3 days ConA/β-ME stimulation) and (b) rMSC + hPBMC (5 days PHA + IL2 stimulation); and allo-cocultures: (c) hMSC + hPBMC (5 days PHA + IL2); (d) rMSC + r PBMC/rSMC, 3 days, ConA/β-ME).

#### 2.5.4. Assessment of PBMC/SMC Proliferation

To assess the anti-proliferative effect of MSC on mitogen-stimulated PBMC/SMC, cells were harvested from coculture and control conditions. Technical replicates were pooled and resuspended in FACS buffer (PBS supplemented with 0.4% BSA and 0.02% NaN_3_). Next to assessing proliferation of PBMC/SMC by Cytotell Green dye dilution, and to compare human whole PBMC population versus CD4+ T cell subsets, cells were first incubated 5 min with 10 µL of FcR blocking reagent (Miltenyi Biotec), and stained with extracellular marker anti-CD4-PE (BD) during 20 min at 4 °C. Cells were then resuspended in FACS buffer with Sytox Red (1:4000 final concentration, Invitrogen Life TechnologiesDarmstadt, Germany) or Sytox Blue (1:2000 final concentration, Invitrogen Life Technologies). Cells were analysed immediately and .fcs files were exported and analysed with FlowJo 10 software (FlowJo, LLC, Ashland, OR, USA).

PBMC/SMC or CD4+ T cell proliferation was assessed via Cytotell green dye dilution using the FlowJo 7 software proliferation tool. Division index (average number of divisions for all cells in culture) was chosen to define PBMC or CD4+ T cell proliferation [31].

### 2.6. Indoleamine 2,3-dioxygenase (IDO) Measurements

MSC were seeded in a 6-well plate at a density of 5000 cells/cm^2^ in RPMI. Cells were allowed to attach and were then treated for 72 h either with IFNγ (10 ng/mL), Tryp (100 µg/mL) or Epac (1 µM). After trypsinisation and washing, MSC were resuspended in PBS, stained with fixable viability dye eF450 (1:4000 final dilution; eBioscience, Frankfurt am Main, Germany) for 30 min at 4 °C. After washing once, cell suspension was incubated with intracellular fixation buffer (eBioscience, Frankfurt am Main, Germany) during 30 min at room temperature (RT). After washing/centrifuging samples twice with 1× permeabilisation buffer (eBioscience, Frankfurt am Main, Germany), cells were stained with IDO-PE (eBioscience, Frankfurt am Main, Germany) in 1× permeabilisation buffer for 30 min. Cells were washed and analysed immediately at BD FACS Canto II (BD Biociences, Heidelberg, Germany). MSC without IFNγ stimulation served as control. Data are represented as MFI after subtraction of MFI’s respective control.

### 2.7. Kynurenine Measurement

Kynurenine was detected in CM from IDO assays or cocultures (human and rat) and controls of stimulated and not stimulated conditions. A total of 100 µL of samples or standard dilutions (50 mM L-Kynurenine; Santa Cruz Biotechnology) were added to 30% trichloroacetic acid (TCA) (Carl Roth GmbH; Karlsruhe, Germany) in a 96-well plate. After 30 min incubation at 50 °C, the plate was centrifuged and 75 µL of supernatants were carefully transferred to a new plate. 4-(dimethylamino)benzaldehyde (sc-202888, Santa Cruz Biotechnology) were added and incubated for 15 min in the dark at RT. Optical density (OD) was determined using a microplate reader (TECAN infinite M200PRO, Tecan Deutschland GmbH; Crailsheim, Germany) at 492 nm emission wavelength. Standard curves were elaborated with GraphPad Prism 7 software (GraphPad Software; San Diego, CA, USA).

### 2.8. Nitrite Measurement

Nitrite (NO_2−_) was detected in CM from IDO assay or cocultures (human and rat) and controls of stimulated and not stimulated conditions. A total of 150 µL of probes or standard dilutions (0.1 µmol/mL sodium nitrite dissolved in RPMI) were mixed with sulfanilamide (both AppliChem) in a 96-well clear plate. Plate was incubated during two minutes before the addition of N-(1-napthyl)-ethylendiamine dihydrochloride (naphtylamine) (Carl Roth GmbH), then incubated for 30 min in the dark at RT. OD of each well was determined using a microplate reader using 542 nm emission wavelength and 620 nm reference wavelength. Standard curves were elaborated with GraphPad Prism 7 software and limit of detection was calculated and applied to measured values.

### 2.9. Statistics

Data are represented as mean ± standard deviation. Significance testing (2-way ANOVA) were performed with GraphPad Prism v.7 software (GraphPad Software). Asterisks depicted at the top of the whiskers and box plots represent significance of the individual condition with respect to their own positive control. *p* < 0.05 is considered statistically significant (* *p* < 0.05, ** *p* < 0.01, *** *p* < 0.001 and **** *p* < 0.0001). Symbol § represents the significance of the individual condition with respect to their positive control (§: *p* < 0.0001).

## 3. Results

### 3.1. Human Adipose Tissue-Derived Stromal Cells suppress Human PBMC Independent of Extracellular Vesicles via Indoleamine 2,3-Dioxygenase Activity

#### 3.1.1. Inhibition of PBMC Proliferation Is Higher in ASC Cocultures, Independent of Their Passage

In order to verify MSC immunomodulatory capacities, direct cocultures with stimulated PBMC were established. PBMC proliferation (or inhibition thereof) was measured by the rate of dye dilution (Cytotell green) and calculated as division index (DI).

PHA stimulated PBMC or CD4 cells (+PHA positive control) had proliferated as denoted by the elevated DI values (dotted line at 1: normalisation of values to positive control). In all conditions, PBMC division index was significantly reduced with respect to the +PHA control (*p* < 0.001), except CB 1:20 ratio (*p* < 0.01) (Figure 1A). However, when comparing the three MSC sources at all tested ratios (1:5, 1:10 and 1.20), ASC appeared to be significantly more immunosuppressive than CB (DI at ratio 1:5 ASC vs. CB, 0.31 vs. 0.52, *p* < 0.001) (Figure 1A) and BM-mediated suppression was only significant in 1:10 ratio (*p* < 0.01)). MSC inhibitory strength was passage independent, as differences amongst P3 and P5 cocultures were not significant (n.s.) (Figure 1B). When comparing whole PBMC and enriched CD4 T cell cocultures, proliferation was similarly suppressed by MSC (n.s.) (Figure 1C), indicating that under the chosen conditions no accessory cells are required to mediate suppressive action.

#### 3.1.2. Tryptophan Addition to Cocultures Abrogates MSC Mediated PBMC Inhibition, Which Is Correlated with an Increase of IDO and Kynurenine Secretion

IDO has been claimed to be amongst the most relevant mechanisms involved in MSC-mediated suppression of T cell proliferation, leading to degradation of tryptophan to kynurenine [9]. IDO is not constitutively expressed in MSC but needs to be induced by IFNγ.

PBMC DI was highly reduced in direct coculture with MSC (-IFNγ condition). Interestingly, IFNγ-priming did not increase MSC suppressive capacities (n.s., 2-way ANOVA, CB and BM not tested here) (Figure 2A). Kynurenine levels were similar in unprimed and IFNγ-primed MSC:PBMC cocultures (Figure 2B). No kynurenine was detected in PBMC, stimulated and not stimulated, nor in MSC monoculture controls. Arguing that tryptophan levels may be critical for lymphocyte proliferation, we added tryptophan to the culture medium. In fact, additional tryptophan led to the abrogation of PBMC inhibition in PHA-stimulated cocultures (BM and ASC: *p* < 0.05 and CB: *p* < 0.001) (Figure 2A). Concomitantly, kynurenine values were significantly increased in coculture CM upon addition of tryptophan (BM, CB and ASC -IFNγ -Tryp vs. -IFNγ +Tryp, *p* < 0.0001, *p* < 0.001 and *p* < 0.0001, respectively) (Figure 2B).

We expected increased suppressive strength of MSC upon IFNγ priming. In fact, MSC were able to produce substantial amounts of IDO only upon IFNγ stimulation (Figure 2C). ASC IDO expression was the most noticeable, (*p* < 0.0001), followed by BM and CB (ASC vs. CB, *p* < 0.0001 and ASC vs. BM, *p* < 0.01)). Tryptophan addition did not alter IDO levels. Kynurenine levels increased in IFNγ-primed MSC and were significantly increased upon tryptophan addition (Figure 2D, *p* < 0.0001). BM presented higher values than MSC from other sources (BM vs. CB, *p* < 0.0001 and BM vs. ASC, *p* < 0.05) (Figure 2D).

In conclusion, we found that in presence of IFNγ, MSC suppressive capacities remained unaltered, and although IDO secretion is highly promoted, kynurenine values remained barely detectable. Tryptophan addition on the other hand, effectively abrogated MSC-mediated PBMC inhibition indicating its critical role in MSC-mediated suppression of T cell proliferation.

#### 3.1.3. Epacadostat Abolishes PBMC Inhibition and Decreases IDO Expression and Kynurenine Secretion

To understand the extent by which MSC inhibitory potential was mediated by IDO enzymatic activity, we added the IDO inhibitor epacadostat (Epac). Once ASC were identified as the more immunosuppressant, ASC were chosen for further experiments. Epac entirely reversed the suppressive action of ASC and led to an overproliferation of PHA-stimulated PBMC (DI—IFNγ: + PHA − Epac = 0.64 and DI of + PHA + Epac = 1.42, *p* < 0.001) (Figure 2E). A similar effect was seen when MSC were primed (+IFNγ: *p* < 0.0001). However, this overproliferation was only seen in PHA-stimulated PBMC whilst Epac had no effect on non-stimulated PBMC proliferation (n.s.).

Upon Epac addition, kynurenine levels decreased beyond the detection limit in both stimulated and non-stimulated and IFNγ-primed and non-primed MSC: PBMC cocultures (*p* < 0.0001) (Figure 2F).

IDO levels significantly increased upon IFNγ priming in ASC (value 12,345 + IFNγ vs. 133 − IFNγ) and further increased upon Tryptophan addition (value 15,440), were significantly lowered upon Epac addition (value 10,207 − Tryp + Epac vs. 9723 + Tryp + Epac; *p* < 0.0001) (Figure 2G). Despite this reduction, IDO levels were comparable to levels with only IFNγ stimulation.

#### 3.1.4. Direct and Transwell Coculture Inhibit the Proliferation of Stimulated PBMC via IDO-Kynurenine Pathway

We have shown that ASC inhibit T cell proliferation via the IDO-kynurenine pathway. We were interested in understanding whether a direct cell-mediated communication is required to induce this effect or whether it functions via paracrine factors or EV. First, we tested the ASC inhibitory potential on stimulated PBMC both in a direct and indirect transwell culture. Mitogen-induced PBMC proliferation was significantly inhibited in both direct and transwell settings (*p* < 0.0001, 2-way ANOVA Figure 3A), independent of IFNγ priming indicating that direct cell-cell interaction is not essential for MSC to exert their immune modulation (n.s.).

As shown before, kynurenine levels were high in PHA-stimulated cocultures both in direct and transwell conditions (*p* < 0.0001, 2-way ANOVA) (Figure 3B). MSC prestimulation with IFNγ showed no difference in kynurenine secretion (n.s).

Next, we tested ASC conditioned media (CM) and extracellular vesicles (EV, derived from primed or non-primed ASC) (Appendix A for detailed information about isolation and results of characterisation) modulatory capacities on stimulated PBMC. CM from non IFNγ-primed ASC and EV from either IFNγ-primed or non-primed ASC had no effect on PBMC inhibition, exerting even a slight overstimulation of their division (n.s.), even when used at high ratios equivalent to 20:1 MSC:PBMC. Only CM from IFNγ-primed MSC induced a significant inhibition of PBMC proliferation (*p* < 0.0001, 2-way ANOVA) (Figure 3C). Epacadostat addition did not restore PBMC proliferation in CM +IFNγ (n.s.) (Figure 3D). Kynurenine values measured in proliferation assay cultures with CM +IFNγ and CM +IFNγ +Epac were in accordance with the proliferation data, observing no changes amongst conditions (102 vs. 94 µM, respectively, n.s.). These data show that (1) inhibitory factors are produced and act independently of direct contact between ASC and stimulated PBMC, (2) they are released upon IFNγ priming into the CM and (3) EV do not mediate this suppressive activity even when derived from IFNγ-primed ASC.

#### 3.1.5. Stimulated PBMC Were Not Inhibited by Conditioned Media (CM) Transferred from a Previous Coculture

With our data suggesting a critical role of the IDO-kynurenine pathway as suppressive effector, we asked whether the transfer of CM from a previous 5 day ASC:PBMC coculture could inhibit a subsequent PBMC proliferation in absence of ASC. CM from these cocultures were diluted 1:2 in new full RPMI to ensure newly added factors to the culture. However, CM derived from a previous culture had no inhibitory effect independent of IFNγ-priming or not (n.s.) (Figure 4). Thus, CM from previous cocultures are not able to suppress PBMC proliferation, whereas CM from freshly +IFNγ stimulated MSC effectively inhibit their proliferation.

#### 3.1.6. Nitrite Levels Were Mainly Undetectable amongst All Conditions, Except for MSC-CM

Our data so far indicated a critical role of the IDO pathway as both tryptophan and Epac addition entirely abolished the human ASC-suppressive effects. Regarding already published data which suggest the role of IDO in human and NO in murine MSC-mediated immunomodulation, we assessed nitrite levels in culture media [12]. Results revealed equally negligible nitrite concentrations in both stimulated (around 0.5 µmol/L, n.s.) and not stimulated PBMC monocultures (data not shown). Nitrite levels remained similarly low in supernatants from PHA-stimulated cocultures. Only CM from MSC monocultures (all three MSC sources tested) had higher nitrite levels (around 2 µmol/L) (all MSC from different sources vs. +PHA coculture condition, *p* < 0.0001; Figure 5A). Accordingly, we found very low nitrite concentrations in both IFNγ primed and not primed human coculture supernatants (n.s.; Figure 5B). +IFNγ-priming slightly increased nitrite values in ASC CM with respect to the control (*p* < 0.0001, Figure 5C). Epac did not change nitrite levels. Subsequently, these data suggest that nitrite secretion is not responsible for human PBMC inhibition.

In summary, by performing direct and transwell MSC:PBMC cocultures, we confirmed that MSC immunosuppressive mechanism acts independent of cell-to-cell contact, but is mediated by soluble factors. We revealed the immunosuppressive strength of IFNγ MSC-CM in inhibiting T cell proliferation. Nevertheless, isolated EV failed to suppress their proliferation, regardless of IFNγ priming of MSC.

### 3.2. MSC Immunosuppressive Effect Is Species-Specific

Finally, we questioned whether human MSC can be immunoregulatory even in a xenogeneic setting and whether this relates to IDO or NO mediation. Thus, allogeneic and xenogeneic (human:rat and rat:human) immunosuppression assays were run. Confirming previous data, hASC inhibited hPBMC proliferation dose-dependently (1:5 and 1:10, *p* < 0.01; 1:20, n.s.; Figure 6A). rMSC, however, revealed a significantly reduced hPBMC inhibitory action, often resulting in an overstimulation, compared to hMSC effect (1:5, 1:10, *p* < 0.001).

Contrariwise, hASC led to an overstimulation of rPBMC and rSMC, whereas rMSC suppressed rPBMC proliferation (*p* < 0.01 at least; Figure 6B). Interestingly, rSMC proliferation appeared rather unaffected by rMSC (n.s., 2-way ANOVA).

These data clearly show that allocultures (human:human and rat:rat) exerted immunosuppression, while PBMC in xenocultures (human:rat and rat:human) were not or only marginally inhibited.

#### Kynurenine Secretion Is Prominently Higher in Human PBMC Immunosuppression Assay Whereas Cultures with Rat PBMC Prompt Nitrite Production Further than SMC

Given the described differences for inhibitory mechanisms in human and murine setting [12], we checked for IDO and NO activity in allo- and xenococultures.

Kyn secretion was high only in PHA-stimulated human allocultures (*p* < 0.0001, Figure 6C) and negligible in ConA-stimulated rPBMC and rSMC settings (n.s.; Figure 6D).

In human allocultures, no elevated nitrite levels were detected. Unexpectedly, nitrite levels in rMSC:hPBMC xenocultures were high, ranging from 1 to 30µmol/L (*p* < 0.0001; Figure 6E), whereas nitrite levels in rPBMC and rSMC allo- and xenococultures were not significantly increased compared to the respective controls (Figure 6F). Overall, rPBMC revealed slightly higher values than rSMC (n.s.).

In contrast with our starting hypothesis, these data showed that xenogeneic rMSC:hPBMC coculture is where the utmost nitrite concentration release were found, diverging from the lower levels found in rat allogeneic coculture condition.

## 4. Discussion

### 4.1. Immunomodulatory Potential of Different Human MSC Sources

MSC have gained immense attraction in immunotherapy, regenerative medicine and tissue engineering. MSC can be isolated from a multitude of tissue sources, but mainly bone marrow, adipose tissue, and birth-associated tissues (e.g., umbilical cord, cord blood, placenta) appear to be essential for clinical translation in immune-mediated conditions [3]. However, only a few studies directly compared the immunomodulatory potency of MSC from different tissue sources.

By directly comparing BM, CB and ASC, we revealed that ASC were the most immunosuppressive, inhibiting PHA-induced PBMC proliferation dose-dependently. Supporting previous findings [32], we observed no decreasing activity in older passages, where cells reach senescence (p3 vs. p5). Ribeiro et al. and Najar et al. described that ASC were the strongest suppressive of T cell activation [33] and inhibition of allogeneic T cell proliferation [34], sustaining our findings. However, others have demonstrated BM to be stronger immunosuppressors over ASC [35]. Contrariwise, additional studies support umbilical cord Wharton’s jelly MSC (WJ-MSC) to possess superior immune strength than ASC, BM or placental-MSC [36]. These differences might be easiest explained by the different culture conditions, e.g., serum sources or batches used, as these may influence MSC behaviour [37]. The presence of additional cells (B cells, NK cells, monocytes) and their interplay within PBMC might have an impact on MSC suppressiveness [30]. However, in our setting comparing the whole PBMC population and enriched CD4+ T cells revealed an analogous immunosuppressive action of MSC on both conditions, which hints towards inhibition of largely CD4 subset within PBMC cells and reduced involvement of other additional cells.

### 4.2. The IDO-Kynurenine Pathway Is Essential for MSC-Mediated T Cell Suppression

It is believed that MSC need licensing or preconditioning to exert their immunomodulatory function. In our hands, IFNγ priming did not increase suppressive potential. We presume that in our coculture setting, PHA stimulation acts on PBMC/CD4+ T cells stimulating IFNγ secretion, occurring at an early timepoint. IFNγ then acts on MSC promoting the secretion of IDO into the CM. IDO converts tryptophan and accumulates kynurenine, which acts on T cells, inhibiting their proliferation. We confirmed that IFNγ addition promoted MSC IDO production and immunomodulatory potential. Hence, ASC had highest IDO expression related with their stronger inhibitory capacity. In a similar study, François et al. [38] described a variation among different donors, where stronger IDO producers revealed more potent in vitro T cell proliferation inhibition. Consequently, elevated IDO also related to the increase in kynurenine secretion observed.

To confirm our hypothesis of tryptophan being a rate-limiting factor in IDO modulatory activity, we tested tryptophan addition. Our results evidenced that the addition of Tryp to MSC monocultures greatly abrogated MSC-mediated PBMC inhibition despite further increased IDO and kynurenine levels, inducing in ASC the highest IDO levels. This suggests IDO-mediated reduction of tryptophan levels to be of utmost importance, rather than increased kynurenine levels. This is supported by previous findings which showed a tryptophan-reversible arrest mid G1 phase in MLR cocultures [39].

The use of the IDO inhibitor Epacadostat further proofed this, as Epac fully neutralised the ASC-inhibitory activity. Whilst IDO levels were fairly unaffected, kynurenine concentrations were utterly diminished confirming that Epac inhibits IDO enzymatic activity, but not its secretion. Our findings showed similarities with previous reports where Epac increased T cell proliferation, suppressed Tregs and increased IFNγ production [40]. Furthermore, studies with other IDO inhibitors (1-methyl tryptophan), also described an increased T-cell proliferation in coculture with either naïve or activated MSC [38,41]. Altogether, we confirmed that IDO-mediated tryptophan depletion is key for immune suppression, supporting IDO as the prevailing immune modulator involved in human T cell inhibition in our setting.

Further support for the important involvement of IDO is data comparing direct and indirect cultures. A comparable inhibitory potential of MSC and kynurenine production occurred in both conditions and also in CM primed with IFNγ [42,43].

Surprisingly, Epac did not manage to abrogate CM +IFNγ PBMC inhibition. Thus, we propose that IDO absolute amounts present in the CM are too high to be inhibited by Epacadostat addition, whilst Epac can inhibit IDO enzymatic activity when kinetically produced in cocultures.

CM from previous MSC:PBMC cultures, which contained IDO and kynurenine, however resulted ineffective in inhibiting freshly added PBMC. These results seemed to reveal a possible exhaustion/consumption of enzymatic IDO activity present in the CM after prolonged cultures. Furthermore, the ratio between Tryp, IDO and kynurenine levels, supplied by fresh medium, may have halved their concentrations by diluting the CM, which may not be enough to be suppressive.

The question remains, why suppressive activity is similar when using primed or non-primed ASC. Given its IFNγ-dependency, it is assumed that T cells upon activation release IFNγ, this acts on MSC to produce IDO, which by depleting Tryp and generating kynurenine acts as T cell suppressor to inhibit further T cell activation and proliferation (Mattar et al. medical doctoral thesis). We observed that upon stimulation, PBMC start to release IFNγ at early time points reaching the maximum at 3 h with significant lower levels after 24 h. IDO expression starts within minutes/hours after IFNγ priming. Thus, differences between pre-primed MSC and MSC primed by the coculture may only be seen when studying early suppressive events, for instance T cell activation rather than their proliferation after 5 days.

We defined the importance of the IDO pathway for MSC-mediated T cell suppression in our settings. Thus, we refrained from investigating other MSC modulatory mechanisms.

### 4.3. Extracellular Vesicles, Even When Derived from Primed ASC, Did Not Suppress T Cell Proliferation

MSC-derived CM and EV have been regarded as an advantageous alternative for cell-free therapy, which have already been successfully applied in murine models, along with an arising use in human clinical studies [17,44]. Our results, however, demonstrated that EV isolated from both IFNγ-primed/not primed MSC and not primed MSC CM did not suppress PBMC proliferation, confirming previous observations from others and us [18,21,22]. Contrarily, there are many evidences of EV successfully inhibiting PBMC proliferation [17,19,45,46]. In fact, Blázquez and colleagues detected in a similar setting, EV ability to suppress CD4 and CD8 T cell proliferation [47]. Serejo et al. even reported the successful PBMC proliferation suppression of both unlicensed and IFNγ-licensed MSC-derived EV [48]. Furthermore, EV have been utilized for clinical applications as immunosuppressants, enhancing repair and differentiation or as therapeutic drug carriers [44,49]. Conversely, the absence or reduction of EV immunomodulatory effects on lymphocyte suppression has also been demonstrated [18,20,21,22], hypothesizing that changes in the microenvironment could potentially influence the biological state and release of EV. Indeed, EV potential effectors of immunomodulation might vary with respect to MSC modulatory mechanisms, or as recently suggested, may be batch-dependent (abstract: https://doi.org/10.1016/j.jcyt.2020.03.055). Given that in our setting IDO appeared as most important, as inhibiting its enzymatic activity entirely abolished suppressive activity, we argued that transferred EV preparations have only low levels of IDO, if at all. Recent findings, however, indicate that IDO protein can be transferred via EV, modulating T cell suppression and Treg induction [50,51]. So, besides the role of IDO, the lack of EV-related T cell suppression in our setting may be subject to other/additional causes:(a)Dose. Currently, there is a lack of consensus when reporting the dose of EV load added to potency assays. While we and others determined our EV amount as equivalents to producer cells [19,21], many others reported either EV protein concentration [18,47], EV particle counts [52], or number of producer cells defined as units per ml [17]. Pachler et al. tested different amounts of EV (3:1, 1:1 and 1:3 EV: PBMC ratio), and showed similar dose-dependent effects as with MSC within the PHA proliferation assay [19]. Similar to our findings using a 20:1 EV:PBMC ratio, Conforti et al. report that EV at ratios 20:1, 50:1 and 100:1 have negligible inhibitory activity compared to their cellular counterparts. Only the EV preparation enriched for TGFβ, gelactin-1, HGF and PGE-2 appeared to have inhibitory activity, albeit non-significantly. Regarding previously published EV study assays based on EV protein concentration or particle counts, our EV amounts were 5 to 50-fold lower, which we, however, attribute to the use of EV-depleted serum, thus reducing the amount of non-MSC-derived EV.(b)Culture condition. Changes in the microenvironment and culture settings could potentially influence the biological state, release and yield of EV. For instance, abrupt alterations in culture conditions such as FBS-EV depletion or shifting to serum-free media prior to CM production [53], have been reported to prompt modifications in cell metabolism [54]. Changes to EV-depleted medium might also modify cells phenotypical profile or reduce cell proliferation [53]. Thus, we postulate that FBS-EV depletion might have modified the nature of isolated EV, rendering them less effective.(c)EV isolation method. Here we used ultracentrifugation (UC)-based method for EV isolation and in our previous study ultracentrifugation plus ultracentrifugation on a sucrose cushion [18]. The authors who reported IDO to be in EV preparations used polyethylene glycol (PEG)-based precipitation [50] and UC-based methods [51]. UC shows inconsistencies in reproducibility of isolation data [29,55], as they can co-purify non EV-associated proteins or aggregate EV of different phenotypes [56]. Utilising a different isolation method could potentially overcome these limitations.(d)EV characterisation method. Different complementary methods are required in order to validate EV size, concentration and typical markers [57]. Furthermore, such characterisation methods could potentially pose limitations and bias the interpretation of EV yield, integrity and/or purity.(e)Functional readout. EV may not have a direct effect on T cell proliferation as suggested by a few authors. Di Trapani et al. show that adding 3 × 10^6^ EV from resting or primed MSC/1 × 10^4^ PBMC, sorted T, B or NK cells only B and NK cells reduced on proliferation of T and B cells but not of PBMC and T cells. Yet, inhibition of T cell proliferation was modulated upon adding certain EV inhibitors, which prompted the authors to postulate an indirect EV effect [22]. Chen et al. for instance report that EV did not suppress concanavalin-A driven T cell proliferation and did not affect IDO activity. However, EV changed the cytokine milieu and increased the ratio of Treg and Th2 cells in sake of Th1 and Th17 cells [23].

To gain further insight, we now initiated a study where we receive EV from collaboration partners, EV isolated from different cell sources, cultivated in different culture conditions, and isolated using different isolation protocols to compare these within our assay and other immunomodulation assays run in different labs.

### 4.4. MSC Immunosuppressive Effect Is Species-Specific

There are increasing data which indicate species-incompatibilities and lack of suppressive activity in xenotransplantation models [26,58,59]. This challenges the value of animal models in identifying molecular mechanisms of action and performing preclinical validation testing of human MSC. Some studies claim MSC to be capable of inhibiting progression of autoimmune diseases and recover immune homeostasis [60], enhance survival of grafts [61] and transplants, conferring immunomodulatory beneficial effects. Nevertheless, other studies prove MSC infusion to render less or no benefit to graft survival. In a transplantation murine model they demonstrated how pre-infused hMSC incited an allograft rejection prior to day 30 post graft insertion [26], which seemed to claim interspecies incompatibility. While our data supported the notion of hMSC not to be suppressive across species, it is clear that solely performing in vitro experimentation leads to open questions. It is our view that this effect might be directly associated with potential species-dependent varying MSC-mediated inhibitory mechanisms [12,62] or to the strong effect of pro-inflammatory micro-environments.

We found successful inhibition of human and rat PBMC by their respective allogeneic MSC, thus demonstrating the relevance of immune compatibility. In rMSC allo-cocultures, we observed a stronger T cell inhibition of rPBMC than of rSMC. These differences may be explained by the differing composition of immune cell subsets in PBMC and spleen. In spleen and tonsils, CD4+ and CD8+ T cell subsets were much lower than in other tissues, exceeded by B cells [63,64].

To confirm previous findings of IDO as predominant mechanism in human MSC-mediated immunosuppression and NO in murine MSC, we analysed kynurenine and nitrite culture levels. Human MSC cultures, in fact, exhibited elevated kynurenine levels, while nitrite levels were low. These data seem to correlate to previous findings where they tested NO levels of analogous coculture settings and saw no changes in culture medium of mixed human PBMC-ASC cultures. In rMSC allo-cocultures, nitrite levels were increased, yet not significantly. Compared to other murine studies that revealed NO production promotion along with a strong suppression of T-cell proliferation as a result of direct MSC and T cell interaction [11], this may be related to timing (SN tested only after 3 days) and/or stimulation with ConA plus β-ME. The high nitrite levels in rMSC:hPBMC cocultures may be explained by a xenoreaction inducing NO in rMSC, but not resulting in immunosuppression. There was a high variance with one hPBMC donor inducing very high levels. It may also reflect a late response not able to overcome ConA-induced proliferation. It may have been better to investigate iNOS expression, but due to the fact that this was a confirmatory test for species-specificity in immunomodulation, we did not analyse the mechanism in more detail. Confirming previous findings, immunomodulatory capacity, at least under defined in vitro conditions, appears to be species-specific and related to differing molecular immunosuppression mechanisms by ASC in our settings.

In summary, we have reported ASC as the strongest MSC exerting their immunosuppression mainly via IDO-kynurenine pathway. Whilst EV lacked suppressive effects in our setting, CM from primed ASC managed to exert analogous effects as their cellular counterpart. In addition, we revealed MSC immunosuppressive effects to be species-specific; thus, preclinical data need to be carefully interpreted, especially when considering a possible translation to clinical field.

## Figures and Tables

**Figure 1 cells-09-02419-f001:**
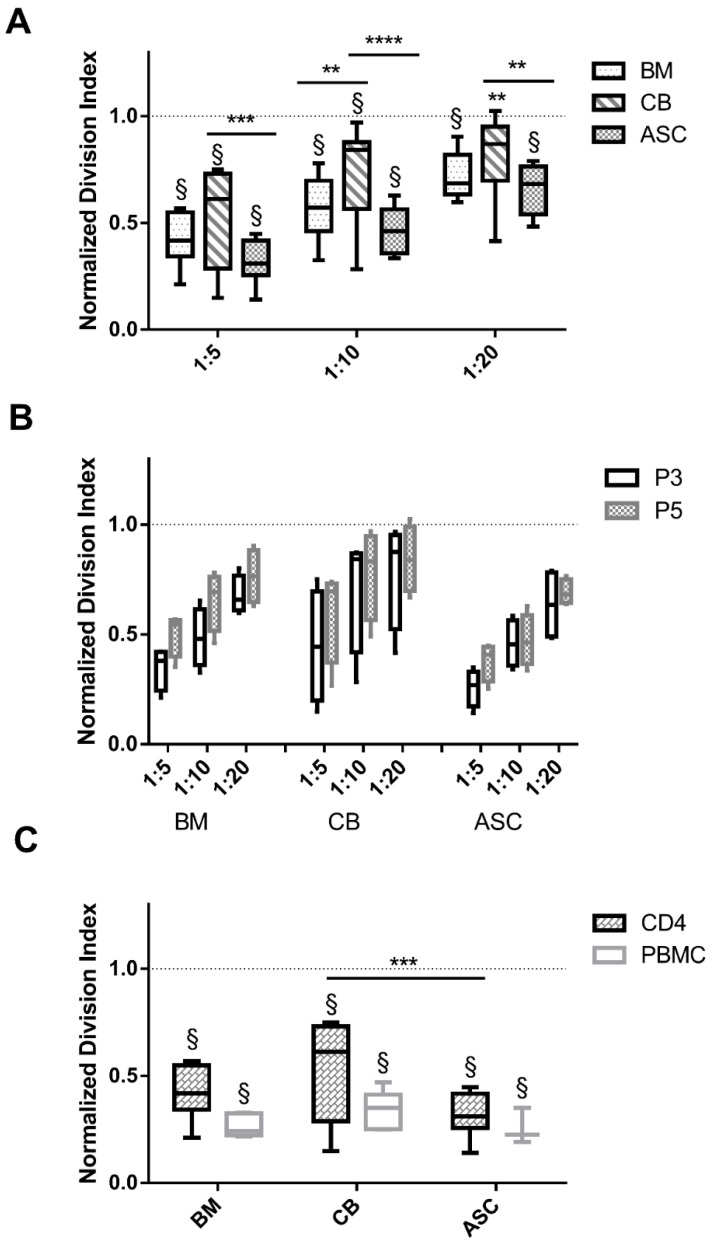
ASC suppress PHA-induced T cell proliferation to a stronger extent than BM or CB-MSC, independent of passage number and CD4 or PBMC population. (**A**) PBMC division index in cocultures with BM, CB and ASC in three different ratios (1:5, 1:10 and 1:20), indicates ASC as stronger immunosuppressors. (**B**) MSC inhibitory potential in P3 and P5 does not differ (n.s., 2-way ANOVA). (**C**) CD4+ T cell and PBMC division index is not impacted (n.s., 2-way ANOVA). Dotted lines represent the normalized division index of the positive control (only PBMC stimulated with PHA: +PHA control). Box: interquartile range; whiskers: minimum to maximum; line: median. Asterisks depicted at the top of the lines represent the significance of the individual value with respect to their own condition control (** *p* < 0.01, *** *p* < 0.001, **** *p* < 0.0001) Symbol § represents the significance of the individual conditions with respect to their +PHA control (§: *p* < 0.0001, 2-way ANOVA). n = 4 to 8, different MSC isolates and different PBMC isolates.

**Figure 2 cells-09-02419-f002:**
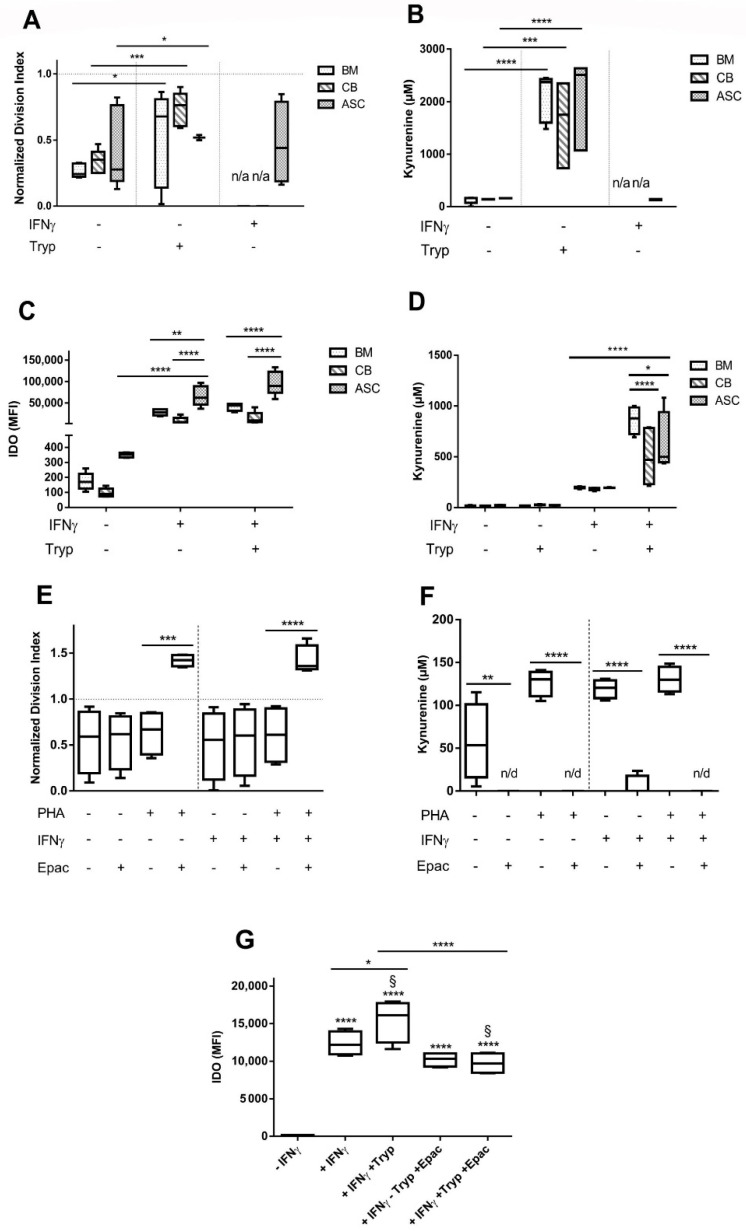
IDO expression is induced by IFNγ stimulation and further increased with the addition of tryptophan, which in turn largely abrogates MSC inhibitory potential. (**A**) MSC:PBMC cocultures were set with BM, CB and ASC cells. Addition of IFNγ and Tryp were tested, n = 5 to 8. (**B**) Kynurenine coculture concentrations measured in presence of IFNγ and Tryp, n= 5 to 8. (**C**) IDO production is significantly increased when MSC are stimulated with IFNγ, n = 5. (**D**) Kynurenine concentrations measured in MSC monocultures after IFNγ addition are increased, n = 5. (**E**) ASC:PBMC cocultures were set with the addition of IDO inhibitor, Epacadostat, n = 4. (**F**) Kynurenine coculture concentrations completely abolished in presence of Epacadostat, n = 4. (**G**) IDO secretion in MSC monocultures is reduced when adding Epacadostat (*p* < 0.0001, 2-way ANOVA), n = 4. Box: interquartile range; whiskers: minimum to maximum; line: median. Dotted lines represent the normalisation referred to the positive control. Asterisks depicted at the top of the lines represent the significance of the individual value with respect to their own condition control (* *p* < 0.05, ** *p* < 0.01, *** *p* < 0.001, **** *p* < 0.0001). Symbol § represents the significance of the individual conditions with respect to their positive control (§: *p* < 0.0001, 2-way ANOVA). Lines with asterisks depict the significance between two conditions.

**Figure 3 cells-09-02419-f003:**
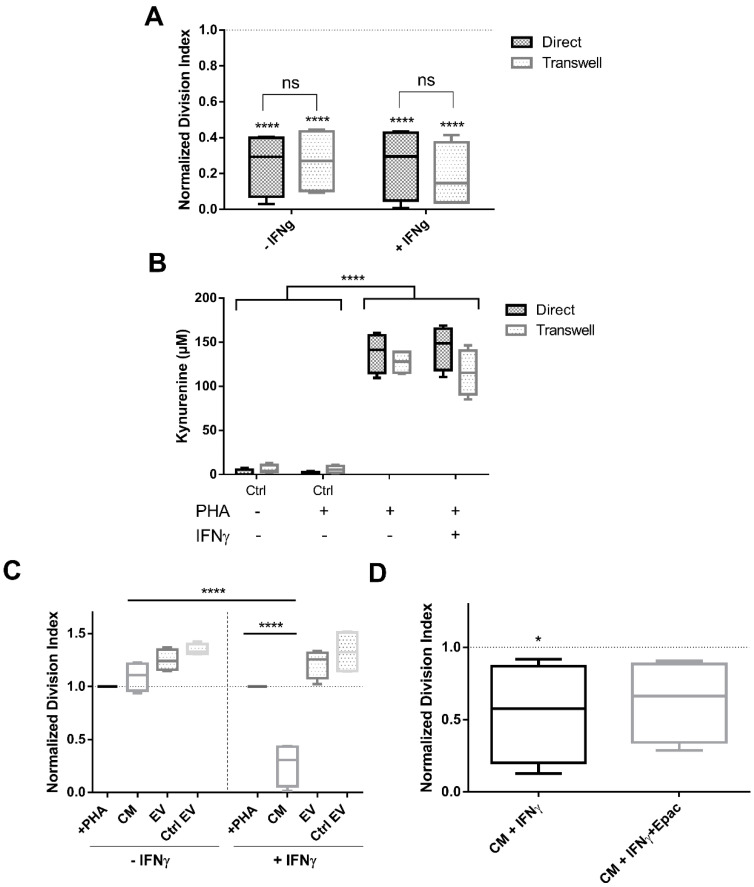
PBMC were equally inhibited in direct and transwell cocultures, although EV failed to suppress their proliferation regardless of IFNγ pre-stimulation. (**A**) PBMC division index evaluation in direct and transwell culture conditions, n = 4. (**B**) Kynurenine concentrations measured in direct and transwell coculture supernatants, n = 4. (**C**) Cocultures with CM and EV isolated from stimulated and not stimulated ASC, n = 4. (**D**) Cultures with CM +IFNγ together with the addition of IDO inhibitor Epacadostat, n = 4. Box: interquartile range; whiskers: minimum to maximum; line: median. Dotted lines represent the normalisation referred to the positive control. Asterisks depicted at the top of the lines represent the significance of the individual value with respect to their own control. Lines with asterisks depict the significance between two conditions (n.s. *p* ≥ 0.05, * *p* < 0.05, **** *p* < 0.0001).

**Figure 4 cells-09-02419-f004:**
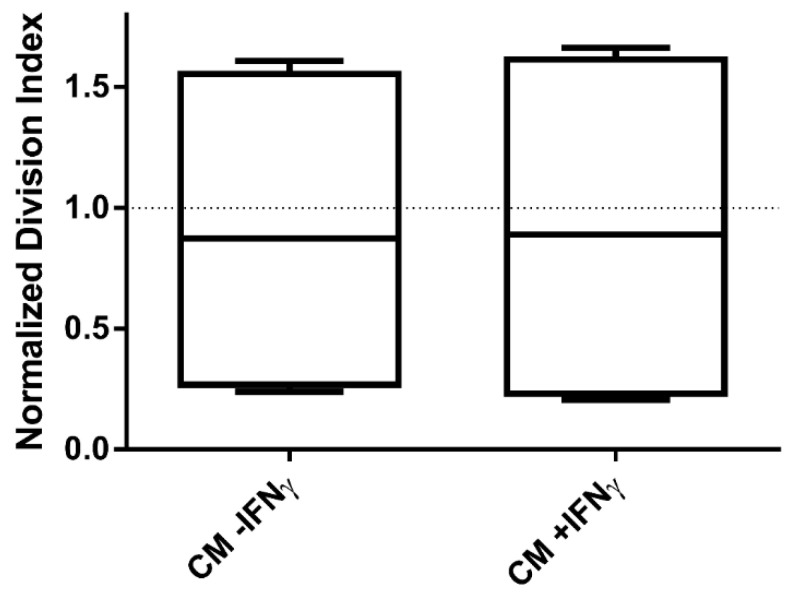
CM transferred from a previous coculture was ineffective in exerting PBMC inhibition. PBMC division index after transferring CM from a 5 day ASC:PBMC coculture (−/+ IFNγ), was analysed. n = 4. Box: interquartile range; whiskers: minimum to maximum; line: median.

**Figure 5 cells-09-02419-f005:**
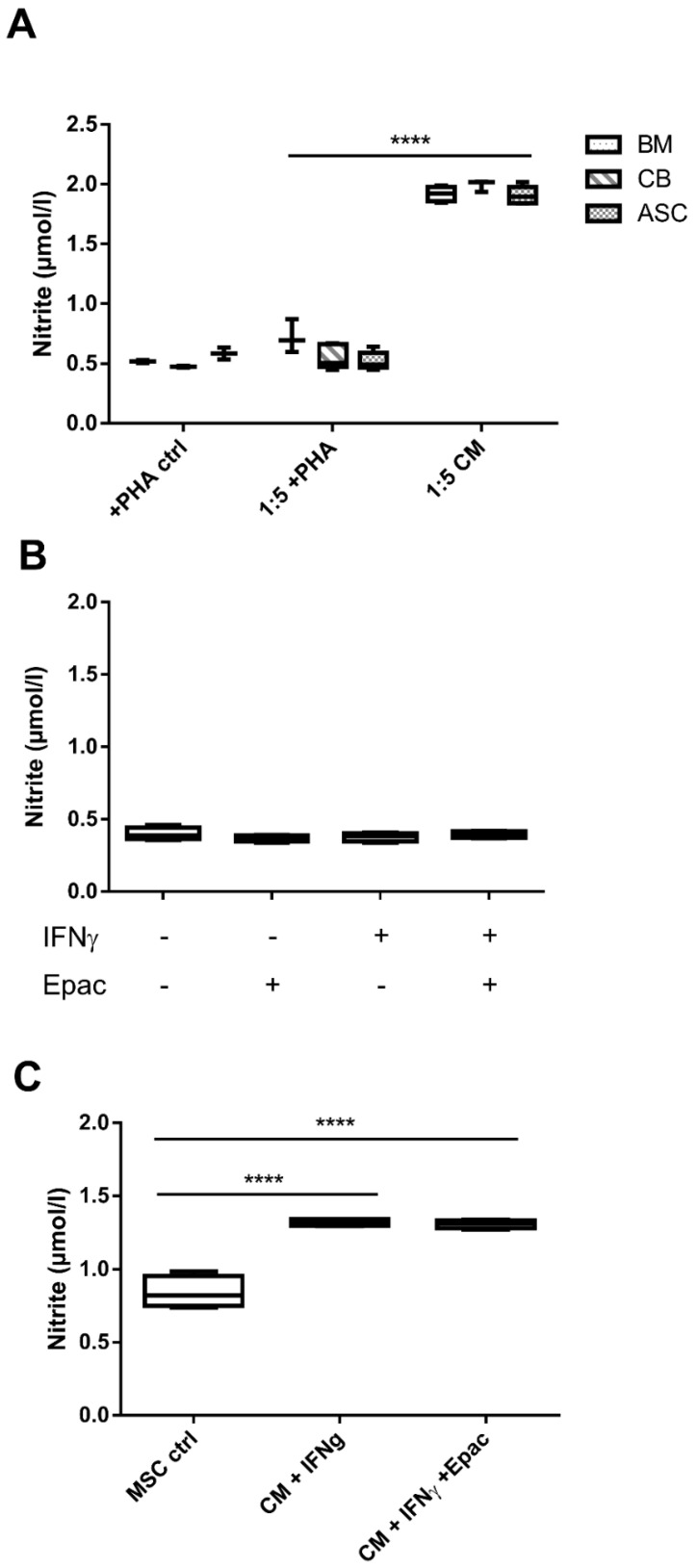
Nitrite concentrations were detectable only in MSC-CM. (**A**) Nitrite concentration levels measured in BM, CB and ASC coculture supernatants. All MSC sources have comparable nitrite production concentration levels, n = 3 to 5. (**B**) Nitrite concentration in coculture supernatants with IFNγ and Epacadostat addition, n = 4. (**C**) Nitrite levels measured in CM +IFNγ after addition of IDO inhibitor Epacadostat, n = 4. Box: interquartile range; whiskers: minimum to maximum; line: median. Lines with asterisks depict the significance between two conditions (**** *p* < 0.0001).

**Figure 6 cells-09-02419-f006:**
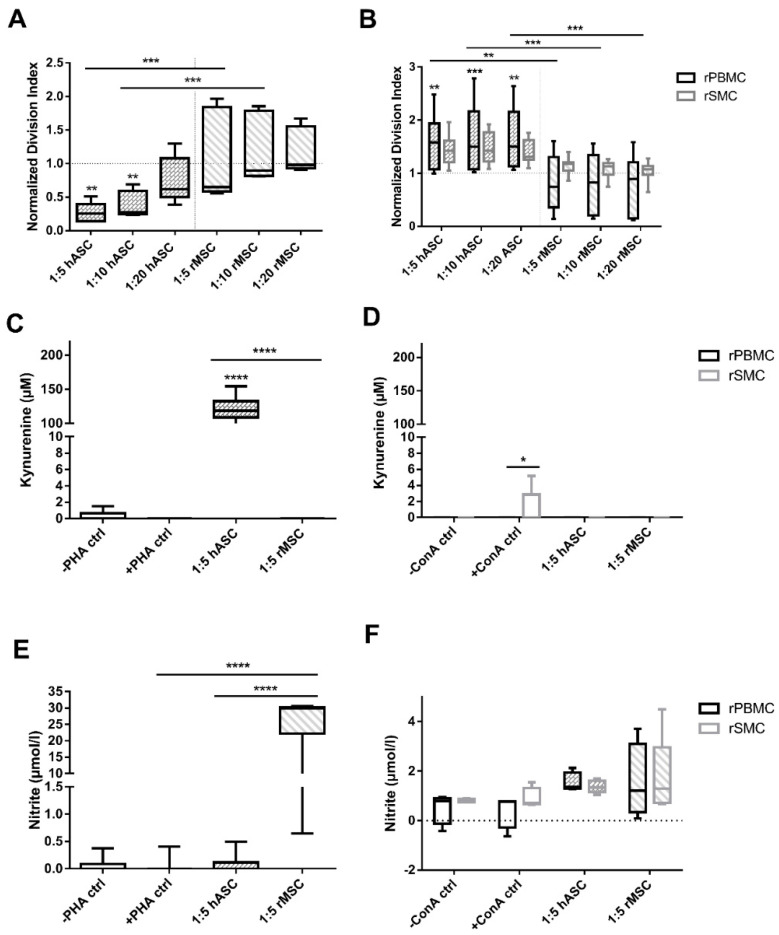
ASC are able to inhibit hPBMC proliferation, but not rPBMC, however, rMSC slightly suppress hPBMC and inhibit blood-derived rPBMC to a higher extent than rSMC. (**A**) Cocultures with hPBMC and hMSC (ASC) and rMSC. ASC inhibited hPBMC in a dose dependent manner, n = 5. (**B**) rPBMC and rSMC cocultures with hMSC and rMS, n = 8 to 10. (**C**) Kynurenine concentrations in hPBMC:ASC or rMSC coculture supernatants, n= 3 to 10. (**D**) Kynurenine concentrations in rPBMC or rSMC:ASC or rMSC coculture supernatants, n = 5 (**E**) Nitrite concentration measured in cocultures with hPBMC, n = 3 to 10. (**F**) Nitrite concentrations in rPBMC or rSMC:ASC or rMSC coculture supernatants, n = 4 to 6. Box: interquartile range; whiskers: minimum to maximum; line: median. Dotted lines represent the normalisation referred to the positive control. Asterisks depicted at the top of the bars represent the significance of the individual value with respect to their own control. Lines with asterisks depict the significance between two conditions. (* *p* < 0.05, ** *p* < 0.01; *** *p* < 0.005, **** *p* < 0.0001).

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
