# Peer review of "Human Adipose Tissue-Derived Stromal Cells Suppress Human, but Not Murine Lymphocyte Proliferation, via Indoleamine 2,3-Dioxygenase Activity"

_cells, 2020, doi:10.3390/cells9112419_

Round 1

Reviewer 1 Report

This manuscript reports the effect of human mesenchymal stromal cells and rodent mesenchymal stromal cells on T-cell proliferation. Additionally explore the mechanism and conclude that the animal model may not be worth as potency assay for human clinical trial.

SUGGESTIONS:

  1. Title- Adipose tissue-derived mesenchyal stromal cells describe better these cells that "Adipose stromal cells"
  2. It i not clear to me where Extracellular Vesicles were derived from. Could the authors include or check dental pulp mesenchymal stromal cells? It is surprising the low effect of EVs
  3. Composition of the Conditioned Media secretome?
  4. Sometimes the conclusions exceed the minimal differences among variables

Reviewer 2 Report

The authors perform an interesting study: I have minor findings:

  • The authors do not describe in results the MSC immunophenotyping of BM, CB and ASC by flow cytometry
  • the authors must implement the discussion by specifically showing the differences between the different MSC funds and above all showing which of the 3 has a better immunosuppressive effect
  • Authors should describe the impact of their study in clinical practice

Reviewer 3 Report

In this manuscript, authors claim that Human adipose stromal cells suppress the proliferation of human peripheral blood mononuclear cells (PBMC, lymphocytes) independent of extracellular vesicles (EVs). Authors claim that EVs lack the ability to suppress PBMCs.

There are several limitations with this study.

This reviewer is afraid if EV researchers can really consider it an EV paper, mainly that EV functionalities are not satisfactory investigated with scientific rigor.

I regret to say that, this is not mainly an EV paper. The whole paper studies ASC- or MSC-mediated inhibition of PBMC proliferation, but not EVs at large. The major drawback of this study is the lack of through investigation for the claims regarding EVs (i.e EVs do not suppress PBMC proliferation). The entire data from Figure 1 to figure 6 is not about EVs. There is minimal data, a minor part which is only figure 3C, that cannot be inferred for the big claim made. Also, there is no method how EV effects on PBMC inhibition were studied (only EV isolation and characterization is presented, but no information about EV assays to study their effect on PBMC proliferation). At functional level, the claims or conclusions should be supported with thorough investigation, and experimental evidence.

The following statement is not true, ‘’there is scarce knowledge regarding EV immunomodulatory properties. Authors might be not familiar with seminal studies by EV experts who have shown, in many reports, the immunomodulatory properties of EVs. In fact, there are hundreds of publications and reproducible data showing the immunomodulatory properties of EVs.

The statement (data not shown), is no more accepted in new publication procedures.

Looking at supplementary figures regarding characterization of extracellular vesicles (EVs), I am afraid what authors have isolated hardly represent EVs, keeping in view the MISEV 2018 guidelines by ISEV society.

Round 2

Reviewer 3 Report

The reviewer agrees with the significance of this study, as is clear from the data itself, however, the major concern remains about inferring the EV part about the conclusion of whole study.

If we consider the title as of its current form i.e. Human adipose tissue-derived stromal cells suppress human, but not murine lymphocyte proliferation, independent of extracellular vesicles…’’

There are two parts of the study, one is about Human adipose tissue-derived stromal cells, and other is about EVs. The first part is very well conducted and holds potential clinical significance, but the data about the roles of EVs on proliferation is very very little to replace the entire conclusion (as authors also agree it is minor part of the study). The point here is entire data set from Figure 1 to figure 6 is about Human adipose tissue-derived stromal cells and their effect on proliferation. Only figure 3C tells little about EV role in this whole context, that cannot reflect the whole data/conclusion from figure 1 to figure 6, or what title says.

Agreeing that authors have very well emphasized the discrepancies in the studies and the discussion has become reasonably convincing, but only part that authors may re-consider is, at least mention the limitation of current study at the end of abstract as well as in the discussion. 

Best wishes 
